# Intolerable Burden of Impetigo in Endemic Settings: A Review of the Current State of Play and Future Directions for Alternative Treatments

**DOI:** 10.3390/antibiotics9120909

**Published:** 2020-12-15

**Authors:** Solomon Abrha, Wubshet Tesfaye, Jackson Thomas

**Affiliations:** 1Faculty of Health, University of Canberra, Canberra, ACT 2617, Australia; Solomon.Bezabh@canberra.edu.au (S.A.); Wubshet.Tesfaye@canberra.edu.au (W.T.); 2Department of Pharmaceutics, School of Pharmacy, College of Health Sciences, Mekelle University, Mekelle 7000, Ethiopia

**Keywords:** review, antibacterial agent, antimicrobial resistance, Australia, hydrogen peroxide, impetigo, manuka oil, tea tree oil, treatment

## Abstract

Impetigo (school sores) is a common superficial bacterial skin infection affecting around 162 million children worldwide, with the highest burden in Australian Aboriginal children. While impetigo itself is treatable, if left untreated, it can lead to life-threatening conditions, such as chronic heart and kidney diseases. Topical antibiotics are often considered the treatment of choice for impetigo, but the clinical efficacy of these treatments is declining at an alarming rate due to the rapid emergence and spread of resistant bacteria. In remote settings in Australia, topical antibiotics are no longer used for impetigo due to the troubling rise of antimicrobial resistance, demanding the use of oral and injectable antibiotic therapies. However, widespread use of these agents not only contributes to existing resistance, but also associated with adverse consequences for individuals and communities. These underscore the urgent need to reinvigorate the antibiotic discovery and alternative impetigo therapies in these settings. This review discusses the current impetigo treatment challenges in endemic settings in Australia and explores potential alternative antimicrobial therapies. The goals are to promote intensified research programs to facilitate effective use of currently available treatments, as well as developing new alternatives for impetigo.

## 1. Introduction

### 1.1. Background

Disabilities secondary to skin conditions are substantial, and skin diseases elicit a huge burden in the global context of health, affecting about 1.9 billion people at any given time [1,2]. In 2017, they were ranked in the top 25 leading causes of Disability Adjusted Life Years (DALYs), accounting for 44.1 million DALYs worldwide [3]. Skin and soft tissue infections (SSTIs) contribute towards 65% of the DALYs of the total skin diseases. SSTIs commonly affect children who live in tropical and subtropical climates, particularly within resource-poor communities, where they affect 21–87% of this age group [1].

Impetigo (school sores), a common superficial skin infection [4], is a huge problem worldwide, affecting >2% of the global population [3], and an estimated 162 million children (particularly those between the age of 2 and 5 years) suffer from impetigo at any one time [5,6]. It is one of eight dermatologic conditions listed in the 50 most common causes of disease in the Global Burden of Disease study [7], and is the only skin condition with potentially life-threatening complications [8]. It is endemic in a number of low-and middle-income tropical countries, with the highest burden (median prevalence 19.4%) in underprivileged children from marginalised communities of high-income countries [5]. In Australia, up to 49% of Aboriginal children living in remote communities are affected by impetigo at any given time (median prevalence of 44.5%) [5], accounting for the highest rate documented anywhere in the world [9]. Given impetigo is considered a disease of the poor (i.e., burden increases as socio-economic status decreases), and its contagious nature [5], inadequate housing, and poor hygiene practices contribute to the growing burden of impetigo in the community [10].

### 1.2. Microbiology of Impetigo

Impetigo is caused by the Gram-positive bacteria *Staphylococcus aureus*, *Streptococcus pyogenes* (group A beta-haemolytic streptococcus, GAS), or a combination of these two bacteria [5,11]. The bacteria may invade the healthy (primary infection) or injured skin (secondary infection), such as atopic dermatitis, insect bites, or scabies, which disrupt the healthy skin barrier and facilitate the entry of pathogenic bacteria into patient’s blood [12,13]. Scabies infected children are 12 times more likely to develop impetigo compared to those with healthy skin [14].

The bacterial aetiology of impetigo varies according to climatic regions, and continues to evolve over time [15]. In tropical climatic regions, GAS is considered as the major pathogen and co-infection with *S. aureus* is common, while *S. aureus* has largely replaced GAS as the predominant pathogen in temperate climates [15,16,17].

Microbiology of skin sore swabs (*n* = 872) taken from 508 Australian indigenous children revealed co-infection with *S. aureus* and *S. pyogenes* in 58% sores [18], whereas 24% and 9% of sores swabs found with isolates of *S. pyogenes* and *S. aureus*, respectively. The study also revealed that 15% of the *S. aureus* isolates were methicillin resistant *S. aureus* (MRSA). A randomised controlled trial (RCT) aimed at investigating oral co-trimoxazole versus intramuscular benzathine benzylpenicillin for impetigo in a highly endemic region in Australia, reported *S. aureus* (81%; *n* = 412), *S. pyogenes* (90%; *n* = 455), and co-infection (74%, *n* = 377), including a total of 32% cases of MRSA isolates from the sores [19]. Another study conducted in a similar setting involved the collection of skin sore swabs from 41 children, revealed the presence of *S. aureus* (81%), *S. pyogenes* (44%), and co-infection of both organisms (37%) [16]. This study also reported that 39% of the *S. aureus* isolates were MRSA.

### 1.3. Clinical Presentation

Impetigo presents in two forms, bullous and non-bullous (sometimes referred to as *impetigo contagiosa*) [4,11]. Bullous impetigo clinically presents with large fragile, flaccid, and fluid filled bullae around the trunk and upper extremities that less readily rupture into thin, brown crusts, and result from exfoliative toxins produced by *S. aureus* species (Figure 1) [4,12,20].

Non-bullous impetigo, the most common form of impetigo, accounts for 70% of cases, and is caused by *S. aureus* and/or GAS [21]. It is typically identified by clinicians as small erythematous vesicles or pustules that rapidly evolve into superficial erosion covered by gold or yellow-coloured crusts, and is usually seen on the face and extremities (Figure 2) [4,12,22].

In general, the clinical appearances of impetigo forms may be dependent on the type of disease causing bacteria or aetiologic agent; however, the histopathology of both forms appear to be similar and mainly characterised by the formation of intraepidermal pustules [24].

### 1.4. Morbidity from Impetigo 

Morbidity of impetigo depends on the bacterial aetiology [15]. In impetigo elicited by *S. aureus*, local and systemic spread of infection could result in cellulitis, lymphangitis, or septicaemia (Figure 3) [13]. Skin infections are the most common signs of initial infection leading to Staphylococcal bacteraemia [25]. In Australia, the 30-day all-cause mortality rate for *S. aureus* sepsis was 16.7% in 2016 and 14.8% in 2017, based on the Antimicrobial use and Resistance (AURA) 2019 report [25]. The annual incidence of invasive *S. aureus* is 10 times higher (46.6 v 4.4 per 100,000 children) in the Indigenous paediatric population compared with the non-Indigenous counterparts living in impetigo endemic settings [26]. *S. aureus* sepsis has a mortality rate of 3–5% in children aged under 18 years, with particularly poor health outcomes in Indigenous children [27]. Similarly, infection with GAS can cause GAS bacteraemia, and Indigenous children (incidence, 69.7 per 100,000 per year) have the highest reported rates of invasive GAS, as opposed to non-Indigenous children (incidence, 8.8 per 100,000 per year) [28]. It can also cause acute post-streptococcal glomerulonephritis (APSGN) and acute rheumatic fever (ARF), inflammatory illnesses that can potentially lead to chronic kidney disease (CKD) and rheumatic heart disease (RHD), respectively [29].

APSGN can develop within 2–6 weeks of untreated streptococcal infection [30], and in Australia, APSGN is linked to streptococcal impetigo in endemic areas, as opposed to pharyngitis in non-endemic areas [31]. A recent study aimed at reviewing and comparing APSGN cases between 2009–2016 in Northern Australia with other countries reported that 94% of the cases found in Indigenous Australians, of which 86% of them were Indigenous children <15 years of age [32]. Indigenous children (incidence, 124.0 per 100,000 per year) had incident rates much higher than non-Indigenous children (incidence, 7.4 per 100,000 per year). The study also showed the incidence of APSGN in children in northern Australia was the highest in the world [32], with a substantially increased risk of progressing to end stage renal disease later in life [30].

The ARF occurs after a latency period of 2–3 weeks post streptococcal infection [30]. Approximately more than 60% of all ARF cases progress to RHD [29], and RHD is estimated to affect at least 2.4 million children (5–14 years) worldwide, with the vast majority (79% of cases) occurring in developing countries and marginalized communities of the developed nations [29]. There is a lack of well-designed studies to establish a clear link between impetigo and ARF in Indigenous communities [33]. However, reports from Australia and New Zealand [34,35,36] indicating high rates of ARF in settings with high incidence of impetigo and low rates of streptococcal pharyngitis could provide some insight into a plausible link between impetigo and ARF and its subsequent progression to RHD. Further, ARF and RHD are highly prevalent among Indigenous communities than other Australians, with the highest rates of ARF found among children aged 5–15 years [37,38,39]. According to the Australian Government Institute of Health and Welfare report [38], Indigenous people are 69 times more likely to develop ARF than their non-Indigenous counterparts and 64 times more likely to have RHD, showing that ARF and RHD almost exclusively affect these communities. Moreover, Indigenous Australians are up to 20 times more likely to die from ARF and RHD as opposed to other Australians [37]. In sum, the long-term consequences of ARF, invasive sepsis, kidney disease, and RHD result in lifelong chronic illness and premature disability in children.

Apart from these potential chronic complications, impetigo is typically characterized by pain, itching, discomfort, and sleep disturbance, substantially impacting the wellbeing of those affected [40]. Due to the contagious nature of the infection, the children are often forced to stay home, excluded from schools or daycares until the infection resolves, and this in turn may require the parents to take time away from work to care for their children [5,41].

## 2. Current Impetigo Treatments and Challenges

Antibiotic therapy is indicated for faster symptom resolution as well as eliminating and/or limiting the spread of the disease from person-to-person [4,17,21]. In Australia, mild and moderate impetigo forms are usually treated with topical antibiotics (i.e., mupirocin) whereas severe or recurrent infections, are treated with oral antibiotics (i.e., dicloxacillin, flucloxacillin, cefalexin, and trimethoprim/sulfamethoxazole [co-trimoxazole]) [42]. Several systematic reviews also suggest topical antibiotics as the treatment of choice for impetigo and systemic antibiotics for complicated impetigo with extensive infections [13,17,43,44]. However, there is a lack of clear conclusive evidence regarding the difference in clinical efficacy between topical and systemic antibiotics for impetigo [13]. A significant number of reports suggest that topical treatments may be superior or equivalent to the oral therapy even for treating extensive form of impetigo [17,21,45,46,47,48].

Unlike systemic antibiotics, topical therapies reduce the potential for systemic absorption and side effects (i.e., gastrointestinal), and also lower the potential for developing resistance to life saving systemic antibiotics [11,13,17,46]. They are also likely to have better compliance, acceptance, and tolerability than systemic antibiotics due to their shorter treatment courses, and ease of application [20,46]. They are suitable for direct application to the infection site, and can deliver high drug concentrations to eradicate the bacteria, thereby reducing the potential of antimicrobial resistance [17,21,49]. These characteristics have therefore made topical treatments the most preferred for mild and moderate impetigo [11,13,17,46].

Like other antibiotics, overuse of topical antibiotics can drive increased antimicrobial resistance (AMR) and rapid emergence of multidrug-resistant bacterial strains (MRSA and macrolide-resistant *S. pyogenes*) is essentially threatening the availability of life saving treatments [16,18,46,50,51]. According to a 2017 WHO report, MRSA is one of the “priority pathogens”, posing substantial risk to human health because they are resistant to most existing treatments [52]. In impetigo, *S. aureus* resistance to fusidic acid, a traditionally used drug for impetigo treatment, has become ubiquitous, potentially limiting its overall efficacy [53,54,55,56]. The emergence of resistance to topical mupirocin, which is normally used as first-line impetigo treatment in different countries [46,57], has also been increasing particularly among MRSA isolates in many parts of the world, suggesting 24–65% high-level resistance to mupirocin in Australia [58], Canada [59], Jamaica [60], New Zealand [61], USA [62], and Trinidad and Tobago [63]. Even though drug resistance to retapamulin, a newly introduced topical treatment for impetigo caused by MSSA and *S. pyogenes*, is thought to be unlikely because of its unique mode of action [20], there is considerable emerging evidence on resistance of MRSA to this drug [64,65]. A recent in vitro study from the United Kingdom investigating XF-73, retapamulin, mupirocin, fusidic acid, daptomycin, and vancomycin against MRSA, indicated that development of drug resistance to retapamulin would be inevitable in the near future [64]. Another in vitro study revealed 10% of the screened isolates showing resistance to retapamulin, of which 57.9% were MRSA [65].

Because of rapidly emerging and spreading antibiotic-resistant bacteria, initial treatment of impetigo with topical antibiotics has been discontinued in remote areas in Australia and, instead, intramuscular (IM) benzathine penicillin G (BPG) alongside oral co-trimoxazole syrup (known by the brand names Bactrim and Septrin) are being employed as the current first-line impetigo treatments [22,27]. The use of oral and injectable antibiotics for uncomplicated impetigo alter systemic levels of host-protective bacteria, and they are also associated with adverse effects (particularly gastrointestinal) [40,66]. In the long run, given the rapid emergence of community-associated methicillin-resistant *S. aureus* (CA-MRSA) in remote areas [67,68], exhausting currently available systemic antibiotics for the treatment of diseases, such as impetigo, can have serious ramifications [40,66,69]. Based on a seven-year descriptive study conducted in New South Wales [70], 33.4% of Aboriginal children were found to have skin infections with CA-MRSA, indicating that CA-MRSA is a significant public health problem in these settings. According to the 2019 Antimicrobial Use and Resistance in Australia (AURA) report [25], about 85–90% of *S. aureus* strains are now resistant to penicillin. Further, about 50% of MRSA are resistant to erythromycin and ciprofloxacin, and 15% are resistant to co-trimoxazole, tetracycline and gentamicin. In a report from the Kimberley region (North Western Australia), one of Australia’s most remote areas, MRSA resistance to co-trimoxazole showed an increase from 9% to 18% over a 12-month period, and following this, the local antimicrobial resistance committee recommended removing co-trimoxazole as the first-line oral alternative for these skin infections [69]. Given the rapid emergence of these resistant bacteria to the current topical antibiotics, a post-antibiotic era is fast approaching, requiring alternative means of treatments with an intension to break the cycle, and prevent further resistance.

In addition to AMR, unsustainable production and supply of existing impetigo antibiotics have been reported as a serious problem in remote settings, limiting the impetigo treatment options for children [71]. Oral co-trimoxazole is available as a syrup suitable for young children under the brand names Bactrim and Septrin. In September 2018, Bactrim syrup was withdrawn from the market, leaving only a single brand of this antibiotic syrup available. As a result, the remaining manufacturer of the antibiotic has been unable to keep up with the demand. Septrin syrup has now been out of stock for over a year, and according to Australia’s Therapeutic Goods Administration Medicine Shortages Information Initiative, it is expected to be unavailable until 2021 [72]. This, in turn, has left the health professionals in the area with the alternative option of crushing and giving the adult dose of co-trimoxazole tablets to children [73]. This practice, not to mention the intolerable taste of the crushed tablets, is not recommended by antibiotic regulators as it does not meet the regulatory standards for administering a dose accurately to young children [73].

Similarly, BPG administration is painful and often leads to poor compliance—this is due to the higher injection volume (1.6–2.3 mL per injection), viscosity (due to high concentration of suspended BPG particles), and irritant nature of the suspension (leading to injection site reaction, such as pain, inflammation, erythema, swelling, and skin ulcer) [39,73,74]. In addition, in case of recurrent impetigo infection, the fear of painful injection may result in needle phobia and non-compliance with the therapy [74]. This shows that the current impetigo treatment options for Indigenous impetigo patients seem far more challenging that it often results in suboptimal clinical outcomes.

To offset the drug shortages and decrease the risk of treatment failures due to AMR, it is important to have multiple impetigo treatment options available. It is also quite clear that antibiotic-resistant bacteria are emerging faster than the pace of replacing the existing treatments with new antimicrobial agents [21,50,58,75,76]. Given the *S. aureus* and GAS species tendency to quickly develop resistance to drugs [11,58,77], they are likely to limit the potency of the current treatments, clearly showing the urgent need for new and effective impetigo treatments options. Ideally, the newer treatment options should have unique modes of action compared to current topical impetigo antibiotics and possess strong activity against *S. aureus* and GAS, as well as resistant isolates [45,49,58].

Given the enormous burden of impetigo in Australian aboriginal children, this review aims to explore the potential of promising alternative selections for impetigo treatment. The candidates selected include tea tree oil (TTO), Manuka oil (MO), and hydrogen peroxide (H_2_O_2_). These are selected because of their potent antibacterial profile against the impetigo-causing bacteria, long-history of their medicinal usage in the community, and their unique modes of action compared to existing impetigo treatments. While this review does not include an exhaustive list of potential alternatives, this may ignite the conversation for aggressive research into finding effective alternative treatments for impetigo.

## 3. Potential Antimicrobial Candidates for Impetigo

### 3.1. Tea Tree Oil

Ever since the multidrug resistance microorganisms appeared as a major medical concern, screening of natural products, in search for new antimicrobial agents has become imperative [78]. Because of their inherent antibacterial, antifungal, antiviral, insecticidal, antioxidant, and anti-inflammatory properties, essential oils obtained from plant materials have traditionally been used for various medicinal purposes including treatment of skin infections, and there has been a growing global interest in their use as substitutes for synthetic antimicrobials [79,80,81,82]. However, most of these essential oils have weak to moderate antimicrobial activities, and they are always overshadowed in practice by more active synthetic agents [80,83]. In fact, only a few of them produce antimicrobial activity against *S. aureus* and *S. pyogenes*, and tea tree oil (TTO), an essential oil obtained from *Melaleuca alternifolia*, is one of these rare essential oils that exhibits potent antibacterial activity against these bacteria [80,83,84,85]. TTO has up to 100 different active compounds—the main constituents are terpinen-4-ol, γ-terpinene, α-terpinene,1,8-cineole, and terpinolene [86,87]. The levels of these components are also specified under an International Organization for Standardization standard (ISO 4730) [86,87]. Even though some authors raised the allergic potential of TTO [88], comprehensive skin sensitivity studies have concluded that TTO is safe for topical application when it is incorporated in a suitable pharmaceutical base at concentrations ≤25%, with no signs of allergic and/or contact skin sensitization [89,90,91,92,93,94,95]. It is also registered in the medicines and healthcare products regulatory agency, UK (THR00240/0399) and Australian Register of Therapeutic Goods (ARTG number:79370) as a herbal medicine [87].

TTO has gained much interest from scientists, physicians and consumers because of its broad-spectrum antimicrobial activity against a variety of bacteria including MRSA, and has long been used as an antibacterial agent for variety of skin conditions by the Aboriginal and mainstream Australian communities for over a century with good safety and efficacy data [87,96]. The activity of TTO against various Gram-positive bacteria has been well documented [86]. It is particularly effective at low concentrations against impetigo causing bacteria (minimum inhibitory concentration (MIC): 0.5–1.25 and minimum bactericidal concentration (MBC): 1–2% (*v*/*v*) vs. *S. aureus*; MIC: 0.04–0.35 and MBC: 0.5% (*v*/*v*) vs. MRSA), and MIC: 0.12–2 and MBC: 0.25–4% (*v*/*v*) vs. *S. pyogenes* [86,87], indicating its promise as a topical antimicrobial agent for impetigo.

The compelling in vitro activity of TTO against MRSA has gained considerable interest and it seems to have translated well with the positive outcomes observed in MRSA colonised patients in randomised controlled clinical trials [89,90]. Caelli et al. (2000) evaluated the clinical efficacy of TTO by randomly allocating MRSA infected patients (*n* = 30) to either routine care (2% mupirocin nasal ointment and triclosan body wash, no report on dose and frequency of administration) or TTO (a 4% tea tree oil nasal ointment and 5% tea tree oil body wash, no report on dose and frequency of administration) given for a minimum of three days [89]. The study reported that more participants were cleared of MRSA carriage in the TTO group (33%) compared to the routine care group (13%) (*p* > 0.05), indicating TTO therapy may be effective in decolonising MRSA carriers. Dryden et al. (2004) also compared MRSA clearing efficacy of TTO regimen (tea tree 10% nasal cream given three times daily plus tea tree 5% body wash given once daily) with standard treatment regimen (mupirocin 2% nasal ointment given three times daily plus chlorhexidine gluconate 4% soap given once daily and silver sulfadiazine 1% cream skin treatments given once daily for five days) in MRSA infected hospitalized patients (*n* = 224) [90]. The study reported that among the participants, 49% in the standard treatment and 41% in the TTO groups were cleared of MRSA carriage, showing no significant difference between the treatment regimens (*p* = 0.0286). TTO treatment was also highly effective at clearing superficial skin lesions compared to the standard treatment (47% versus 31%, respectively, no report on *p* values), indicating its potential use for MRSA-implicated skin infections, such as impetigo.

TTO has also demonstrated a potent anti-biofilm activity against MRSA biofilms with pooled MIC and MBC data of 0.125–2% and 1–8% (*v*/*v*), respectively [97,98]. It also completely eradicated MRSA biofilm cultured from an infected cochlear implant within an hour in one in vitro study [99], and was as effective as vancomycin in eradicating MRSA biofilm on tympanostomy tube in another in vitro study [100]. Biofilms are a complex and organized bacterial communities that are embedded in a self-produced polymeric matrix, which could prevent the antibiotics from entering and become in contact with the bacteria [101]. These bacteria within biofilms may be 100–1000 times less susceptible than their free-living counter-parts [101]. Additional to its antibacterial effects, TTO is effective at low concentration as an anti-inflammatory agent (≤0.125%) that could offer additional benefits when it is used for treatment of skin infections [87].

The mechanism by which TTO produces antibacterial action has not been fully elucidated. Evidence, however, show that treatment with TTO and/or its bioactive components may compromise the cytoplasmic membrane integrity of the bacteria, resulting in the leakage of cell contents, and subsequently disrupts the cellular homeostasis, progressing to cell death [86,102]. This is attributed to the synergistic effect stemming from >100 bioactive components of TTO. This may reduce the potential for developing resistance to TTO, as multiple simultaneous mutations would be required to overcome all the actions of the individual components [86,102,103,104].

Despite the solid in vitro antimicrobial data suggesting its immense potential for treating superficial skin infections including impetigo, TTO is yet to be explored in RCTs for impetigo [105]. Given its potent antibacterial activity against impetigo-causing bacteria, excellent clinical safety profile, and minimal chance for developing resistance, TTO is a noteworthy candidate for treating impetigo.

### 3.2. Manuka Oil

Manuka oil (MO) or *Leptospermum scoparium* oil or Manuka myrtle, is another essential oil with a long history of medicinal use in the community as a herbal medicine, particularly in New Zealand and Australia [106,107,108]. It is obtained from the leaves and seed capsules of Manuka tree (*Leptospermum scoparium*), an indigenous “tea-tree” native to eastern Australia and New Zealand. In New Zealand, various parts of the plant have been employed in Maori remedies for centuries, particularly as a skin antiseptic, an analgesic and wound dressing application—but the plant is most valued for its essential oil, i.e., MO [106,107,109]. MO is mainly composed of monoterpenes, sesquiterpenes, and triketones [110,111]. The triketones in MO make the oil unique, attributing to its potent antimicrobial activity against Gram-positive bacteria, including antibiotic-resistant strains [107,112,113,114]. MO is currently listed as a complementary medicine by therapeutic good administration (TGA) in the forms of balm (ARTG ID 331181) and cream (ARTG ID 331980) for skin applications. It is also used as a bioactive ingredient in various cosmetic products and herbal medicines.

MO possesses strong antimicrobial activity against impetigo causing bacteria with the MIC: 0.05–0.625% (*v*/*v*) and MBC: 0.25–1.25% (*v*/*v*) vs. *S. aureus* [109,114,115,116,117,118,119,120];. MIC: 0.05–0.12% and MBC:0.12% vs. MRSA [109,114,117,121,122]; and MIC: 1 mg/mL vs *S. pyogenes* [120]. Apart from its antibacterial effects, MO possesses good anti-inflammatory and antioxidant properties that could potentially offer value when treating skin infections [108,123,124].

The actual mechanism of action for MO has not yet been identified but evidence (Alnaimat M. S., 2015, unpublished data) [122] suggests cytoplasmic membrane as a primary target in Gram-positive bacteria basing on the marked cellular lysis observed in MO treated MRSA cells as opposed to the untreated MRSA cells. Compared with TTO, MO has not been extensively investigated for impetigo causing bacteria particularly for *S. pyogenes*, but the available preliminary evidence and its long history of medicinal use for the treatment of skin infections warrants exploring appropriate MO-based formulations for their potential usefulness in impetigo treatment.

### 3.3. Hydrogen Peroxide

Hydrogen peroxide (H_2_O_2_) is a well-known antiseptic agent and has been used for treating skin and wound infections [125]. It has shown to be a potent antiseptic agent against several microorganisms, including impetigo-causing bacteria in the range of 3–30% (0.8 to 8M) concentrations [126,127], but a concentration rage of 1–5% is recommended as safe for human topical use [125].

Although the conclusive MIC and MBC data of H_2_O_2_ against impetigo-causing bacteria is not readily available, possibly attributed to the lack of standardized antimicrobial activity test for H_2_O_2_ and topical antiseptics in general [125]. However, this agent has been explored in a randomized controlled trial involving impetigo patients. Christensen and Anehus (1994) [128] examined the efficacy of topical H_2_O_2_ (1%, *v*/*w*) cream in comparison with topical fusidic acid (2%, w/w) cream/gel, given 2–3 times daily for 3 weeks in non-bullous impetigo patients (*n* = 256, 78% of them had *S. aureus*, whereas 8% found with *S. pyogenes* and 14 % with co-infection). After three weeks, 72% (92/128) of the patients in H_2_O_2_ group were healed compared with 82% (105/128) in fusidic acid group (95%CI for odds ratio: 0.604–1.271), demonstrating H_2_O_2_ was not inferior to fusidic acid, which could promote its efficacy for impetigo treatment. Despite lack of conclusive evidence to support the use of topical antiseptics for impetigo [13], guidelines in New Zealand and UK have recently recommended the use of topical H_2_O_2_ as an initial treatment for localised non-bullous impetigo [129]. The results from a Phase IV randomised controlled trial (*n* = 480) [130] exploring the efficacy of H_2_O_2_, 1% cream for mild impetigo treatment is likely to provide additional insight into the utility of H_2_O_2_ for impetigo treatment.

The mechanisms of action of H_2_O_2_ is not completely understood but it is attributed to the irreversible oxidative damages it could inflict on bacterial membranes and DNA [125,127,131]. Evidence indicates that resistance to H_2_O_2_ has not yet been reported [125]. However, this should be noted in light of limited antimicrobial susceptibility data on H_2_O_2_.

In sum, given its promising clinical efficacy and safety profile, topical H_2_O_2_ treatment deserves more attention in terms of further exploration as a potential first-line impetigo treatment.

## 4. Summary

Impetigo is a bacterial skin infection commonly seen in children. Globally, more than 162 million children suffer from impetigo at any one time. In Australia, impetigo affects up to 49% of Aboriginal children living in remote communities at any one time, making it the highest documented prevalence anywhere in the world. This infection could lead to potentially life-threatening conditions including invasive bacterial infection, chronic heart, and kidney diseases. In addition to these chronic complications, impetigo causes pain, itching, discomfort, and sleep disturbance, substantially impacting the wellbeing of those affected. Topical antimicrobials are typically used as first-line treatment options for impetigo. However, the troubling rise of AMR poses a serious challenge to topical antibiotics, almost inevitably requiring the use of life saving systemic antibiotics. This is particularly evident in endemic settings in Australia—and this approach would have enormous societal and clinical consequences. Hence, there is a critical need to explore safe and effective alternative antimicrobials for topical applications to disrupt the rise of AMR so that lifesaving systemic antibiotics can be persevered for life threatening complications. Considering their potent antibacterial activity against impetigo-causing bacteria, topical TTO, MO, and H_2_O_2_ treatments, warrant further investigations.

## Figures and Tables

**Figure 1 antibiotics-09-00909-f001:**
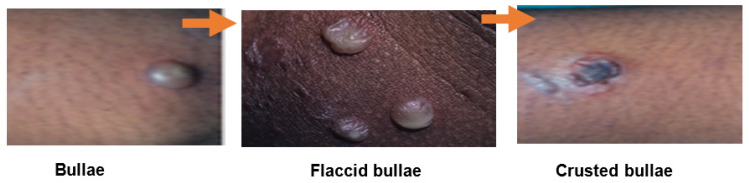
Disease progression of the bullous impetigo (adapted from Cole and Gazewood (2007) [4], and Pereira (2014) [11]).

**Figure 2 antibiotics-09-00909-f002:**
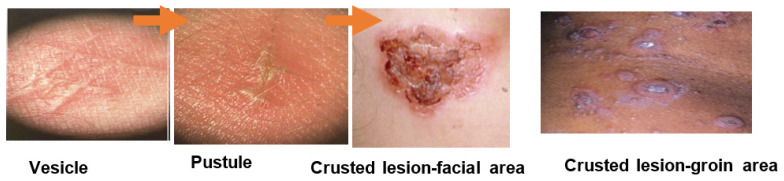
Disease progression of the non-bullous impetigo (adapted from Cole and Gazewood (2007) [4], Pereira (2014) [11], and Leyden et al. (1980) [23]).

**Figure 3 antibiotics-09-00909-f003:**
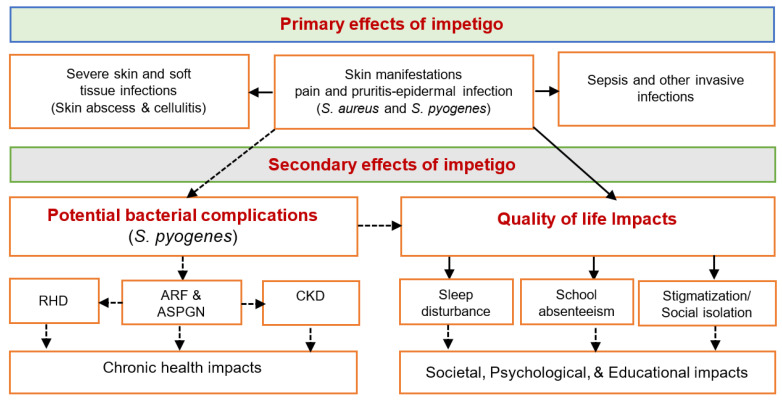
Complications of impetigo (solid arrows (→): direct effects and dashed arrows (⇢): potential secondary effects. Abbreviation: APSGN: acute post-streptococcal glomerulonephritis; ARF: acute rheumatic fever; RHD rheumatic heart disease; and CKD, chronic kidney disease.

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
