# Peer review of "Intolerable Burden of Impetigo in Endemic Settings: A Review of the Current State of Play and Future Directions for Alternative Treatments"

_antibiotics, 2020, doi:10.3390/antibiotics9120909_

Round 1
Reviewer 1 Report
Nice work. Hope my comments for the corrections and revisions are easy to follow.
I just felt that there are a lot of repetitive sections that can be eliminated.
Antibiotics
Title: Intolerable burden of impetigo in endemic settings: a review of “state-of-play” and future directions for research on alternative treatments.
Manuscript ID: Antibiotics: 1028481
General – Interesting topic, enjoyed reading and learning more about the prevalence in different culture. I have listed several suggestions for the author’s consideration. The main issue is some repetition and terminology. Where possible, I gave examples to help with the wording. Also, some sections you talk about “skin diseases” vs. Skin infections, vs. skin conditions. I think it should all be “skin infections”. In some areas, skin condition may be applicable as well. Not sure if “skin disease” is appropriate.
TITLE:
What do the authors mean by “state-of-play”. Not sure if it will be better to just say “a review of the current and future directions for impetigo management”.
ABSTRACT
Line #19: Delete “more”, not necessary, life threatening is strong enough for the condition.
Line #21: Delete “option”, don’t think necessary.
Line #22: Delete “antibiotic” in the antibiotic-resistance bacteria”; just say “spread of resistant bacteria”.
Line#22 & Line#27: Here you repeat “treat impetigo” and “antimicrobial resistance” several time. This section can be consolidated and shortened to avoid repetitive words.
Line #27: Not sure if “rapid” is necessary. If so, you may want to consider replacing it with “brief” or “quick”.
Line #28: end the sentence with: explores potential alternative antimicrobial therapies.
Line #30: When referring to ‘tools” is this meaning the current therapies vs. developing the new ones? Clarify the last sentence.
INTRODUCTION
Background
Line #41: Change “overwhelmingly” to ‘commonly”.
Line #50 – Line #53: I wonder if this can be revised to: in northern Australia, up to 49% of Aboriginal children living in remote communities are affected by impetigo at any given time (median prevalence of 44.5%) accounting for the highest rate documented anywhere in the world.
Microbiology of Impetigo
Line #58 -Line #59: since impetigo may be cause by both organisms, then you may want to revise the sentence to say, impetigo is caused by gram positive bacteria staph aureus, strep pyogenes, or combination of two”.
Line 59 - Line #60: Revise to: the bacteria may invade the healthy skin (primary infection) or through injured skin (secondary infection) such as atopic dermatitis, insect bites…. This will eliminate the repetitive bacteria and bacterial terms.
Line #60: Delete “a” in the (a primary infection)
Line #63: Change “subjects” to “those”
I also think Line #83 – Line #98 fits better under this title since we are discussing the difference in the microbiology and the regions where impetigo is prevalent.
I also would suggest to start a new section titled Clinical presentation. (Line #64 – Line#82)
Line #64: Delete the “forms” after non-bullous.
Line #74: I don’t think it is necessary to repeat the causative organism since you have already reported the common bacteria/organism causing impetigo under microbiology of impetigo.
You discuss the sites where non-bullous impetigo is commonly seen, if know, consider reporting the sites on bullous impetigo. In addition, it sounds like the non-bullous impetigo is classifies as primary and secondary, please verify.
Line #83: Needs clarification, I think the presentation should discuss the Bullous and non-bullous
Line #84: Correct the wording “both forms appear to be similar”.
Line #98: Delete “isolates” at the end of the sentence. You have mentioned this.
Impetigo Morbidity
Line #100: Should this say “morbidity from impetigo, instead “of”?
Line #111 & Line #124: Correct “ASPGN” to “APSGN”
Line #129: Delete “more than 17 times” and replace with “much higher”, the numbers in the parenthesis clarify the numbers.
Line #133: Delete the second “after” and replace with 2-3 weeks post streptococcal infection.
Line #136 – Line #140: Long sentence and confusing. You report that well-designed studies are yet requires, but then mention about reports indicating high rates: where are these reports coming from?
Line #143 and Line #144: Not sure if the exact numbers of 67 times or 64 times is necessary, I think you can just say more likely to develop ARF or have RHD than…..
Line #149: Change “found” to “characterized” by pain….
Line #150: Change “lack of sleep” to “sleep disturbances” impacting the wellbeing of those affected.
Line #151 – Line #154: Should this say “due to the contagious nature of this infection, often the children are forced to stay home from school or daycare until the infection resolves. This in turn may require that the parents have to take time away from work to care for the children.
CURRENT IMPETIGO TREATMENT & CHALLENGES
Line #156 & Line #157: Two short sentences which can be combined for better flow. Here is a suggestion. “Antibiotic therapy is indicated for faster symptom resolution as well as eliminating and/or limiting the spread of the disease from person-to-person”.
Line #157 – Line #160: Just like above, could be shorter and less repetitive. In Australia, mild or moderate impetigo is usually treated with topical antibiotics (i.e., mupirocin) whereas severe or recurrent infections are treated with oral antibiotics (i.e., …….).
Line #161 – Line #167: Change and simplify to, “several systematic reviews suggest topical therapy for first line and systemic therapy for extensive infection. However, there is a lack of clear conclusive evidence between the efficacy of both treatments. Further, a significant number of reports suggest that topical treatments may be superior or even equivalent to oral therapy even for the treatment of extensive impetigo infection.
Line #168 & Line#169: Delete the second “systemic” just say side effects, in addition, delete the whole content in the () and just say i.e. Gastrointestinal.
Line #173: Delete “at the site of action”, you just said they are applied to the infection site.
Line #175: Again, you have two “treatments” here, delete one.
Line #176 – Line #180: Needs revision and cleaning. Not sure how to help. Very confusing and repetitive.
Line #181: The word “priority” does not fit here. I think “widespread” pathogen may be better.
Line #193: Is this supposed to be in the near future?
Line #198: Change “front-line” to “first-line”.
Line #200: Change “adverse consequences” to “adverse effects” as above to keep it consistent. Delete the side effects in the ().
Line #201 – Line #204: Just avoid too much wording and say “In addition, given the rapid emergence of CA-MRSA in remote areas,
Line #204 & Line #205: delete this sentence not necessary. You have a nice section following with the “based on a 7-year…..
Line #215: What is “post-antibiotic world is fast becoming a reality” mean?
Line #233 – Line #234: A different way of reporting will be, this in turn requires using the co-trimazole tablets to compound the syrup which not only has intolerable taste (I thought for pediatric compounding various sweetening agents are available) but also does not meet the regulatory standards specially for accurately administering to children. “Obnoxious taste” is not appropriate vocabulary here.
Line #236 – Line #241: Similarly, with BPG suspension there are several limitations: first, the volume of injection (); second, the viscosity (); third, irritant nature (), results in painful administration which often leads to poor compliance.
Line #241 – Line #242: In addition, in the case of recurrent impetigo infection, the fear of painful injection may result in needle phobia and non-compliance with therapy.
Line #244: Consider this change: often results not only in poor compliance but also suboptimal clinical outcome”.
Line #246-Line #252: Impetigo repeated four times. Please consider deleting the one on Line #250.
Line #255: Change “alternative antimicrobial” to “alternative selection”. Delete the candidates.
POTENTIAL ANTIMICROBIAL CANDIATES FOR IMPETIGO
Tee Tree oil
Line #265: Delete “of” before microorganisms.
Line #269: Replace “in the use of essential oils” with “interest in their use as substitutes for...
Line #273: “Exhibit” should be ”exhibits”.
Line #278: Change “skin application” to “topical application.
Line #283: Change “admiration” to “interest”.
Line #288: Not sure if I saw the definition of “MBC”. Please define ”minimum bactericidal concentration”?
Line #291: Replace “attracted” to “gained”.
Line #294: Revise to, allocating MRSA infected patients (n=30) to either routine care () or TTO (). Delete the words “group” not needed.
Line #305: Delete “received the treatments”.
Line #307: 47% vs. 31% (Vs?)
Line #319: Consider changing “skin diseases” to Skin infections”.
Manuka Oil
Line #338: Delete “and for” and say “an analgesic, and wound….
Line #344: Delete “s” from cosmetic. It should just be “cosmetic products”.
Line # 349: Change “skin disease” to “skin infections or skin conditions” for consistency.
Hydrogen Peroxide
Line #381: Change “frontline” to “first line”.
SUMMARY
“Impetigo” has been repeated multiple times in this
Line 384 and Line #388 – The term “frontline” is not appropriate for the content. Use first-line as it has been listed in previous section(s) of the manuscript.
Line #385: The term “demanding” does not flow here. Consider “requiring”
Line #386: What do you mean by “high-density settings”?
Line #389: Delete “precious”, not necessary and adds no meaning…
Line #392: Financial support: change “is” to “was”.
FIGURES:
Not sure if Figure 4 is necessary
REFERENCES:
No comments, hope it is appropriate per Journals guidelines.
Author Response
Reviewer #1
GENERAL COMMENT
[Comment*]: Nice work. Hope my comments for the corrections and revisions are easy to follow. I just felt that there are a lot of repetitive sections that can be eliminated. Interesting topic, enjoyed reading and learning more about the prevalence in different culture. I have listed several suggestions for the author’s consideration. The main issue is some repetition and terminology. Where possible, I gave examples to help with the wording. Also, some sections you talk about “skin diseases” vs. Skin infections, vs. skin conditions. I think it should all be “skin infections”. In some areas, skin condition may be applicable as well. Not sure if “skin disease” is appropriate.
[#Response#]: We thank reviewer #1 for the kind words and constructive feedback.
Minor comments
TITLE:
- [Comment*]: What do the authors mean by “state-of-play”. Not sure if it will be better to just say “a review of the current and future directions for impetigo management”.
[#Response#]: Thanks, and we used the terms ‘state-of-paly’ to refer to the ‘status or situation’ of impetigo burden and treatment. We have now slightly modified the title as “Intolerable burden of impetigo in endemic settings: A review of the current state of play and future directions for alternative treatments”. (Please see Page #1 and Line #1–3)
ABSTRACT
- [Comment*]: Line #19: Delete “more”, not necessary, life threatening is strong enough for the condition.
[#Response#]: Thanks, and the word “more” is now deleted. (Please see Page #1 and Line #19)
- [Comment*]: Line #21: Delete “option”, don’t think necessary.
[#Response#]: Thanks, and the word “option” is now deleted. (Please see Page #1 and Line #21)
- [Comment*]: Line #22: Delete “antibiotic” in the antibiotic-resistance bacteria”; just say “spread of resistant bacteria”.
[#Response#]: Thanks, and the word “antibiotics” is now deleted. (Please see Page #1 and Line #22)
- [Comment*]: Line#22 & Line#27: Here you repeat “treat impetigo” and “antimicrobial resistance” several time. This section can be consolidated and shortened to avoid repetitive words.
[#Response#]: Thanks, and this part is now modified as “Topical antibiotics are often considered the treatment of choice for impetigo, but the clinical efficacy of these treatments is declining at an alarming rate due to the rapid emergence and spread of resistant bacteria. In remote settings in Australia, topical antibiotics are no longer used for impetigo due to the troubling rise of antimicrobial resistance, demanding the use of oral and injectable antibiotic therapies. However, widespread use of these agents not only contributes to existing resistance but also associated with adverse consequences for individuals and communities. These underscore the urgent need to reinvigorate the antibiotic discovery and alternative impetigo therapies in these settings”. (Please see Page #1 and Line #20–27)
- [Comment*]: Line #27: Not sure if “rapid” is necessary. If so, you may want to consider replacing it with “brief” or “quick”.
[#Response#]: Thanks, and the word “rapid” is now deleted. (Please see Page #1 and Line #27)
- [Comment*]: Line #28: end the sentence with: explores potential alternative antimicrobial therapies.
[#Response#]: Thanks, and this part has now been modified as suggested. (Please see Page #1 and Line #29)
- [Comment*]: Line #30: When referring to ‘tools” is this meaning the current therapies vs. developing the new ones? Clarify the last sentence.
[#Response#]: Thanks, and the last sentence has now been modified as “The goals are to promote intensified research program to facilitate effective use of currently available treatments as well as developing new alternatives for impetigo”. (Please see Page #1 and Line #29–30)
INTRODUCTION
Background
- [Comment*]: Line #41: Change “overwhelmingly” to ‘commonly”.
[#Response#]: Thanks, and the word "overwhelmingly” is now replaced with “commonly”. (Please see Page #1 and Line #40)
- [Comment*]: Line #50 – Line #53: I wonder if this can be revised to: in northern Australia, up to 49% of Aboriginal children living in remote communities are affected by impetigo at any given time (median prevalence of 44.5%) accounting for the highest rate documented anywhere in the world.
[#Response#]: Thanks, and this part is now modified as “In Australia, up to 49% of Aboriginal children living in remote communities are affected by impetigo at any given time (median prevalence of 44·5%), accounting for the highest rate documented anywhere in the world [9]”. (Please see Page #2 and Line #50–52)
Microbiology of Impetigo
- [Comment*]: Line #58 -Line #59: since impetigo may be cause by both organisms, then you may want to revise the sentence to say, impetigo is caused by gram positive bacteria staph aureus, strep pyogenes, or combination of two”.
[#Response#]: Thanks, and this part is now modified as “Impetigo is caused by the gram-positive bacteria Staphylococcus aureus, Streptococcus pyogenes (group A beta-haemolytic streptococcus, GAS) or a combination of these two bacteria [5, 11]”. (Please see Page #2 and Line #57–58)
- [Comment*]: Line 59 - Line #60: Revise to: the bacteria may invade the healthy skin (primary infection) or through injured skin (secondary infection) such as atopic dermatitis, insect bites…. This will eliminate the repetitive bacteria and bacterial terms.
[#Response#]: Thanks, and this part is now modified as “The bacteria may invade the healthy (primary infection) or injured skin (secondary infection) such as atopic dermatitis, insect bites or scabies, which disrupt the healthy skin barrier and facilitate the entry of pathogenic bacteria into patient’s blood [12, 13]”. (Please see Page #2 and Line #58–61)
- [Comment*]: Line #60: Delete “a” in the (a primary infection)
[#Response#]: Thanks, and the alphabet “a” is now deleted. (Please see Page #2 and Line #59)
- [Comment*]: Line #63: Change “subjects” to “those”
[#Response#]: Thanks, and the word "subject” is now replaced with “those”. (Please see Page #2 and Line #62)
- [Comment*]: I also think Line #83 – Line #98 fits better under this title since we are discussing the difference in the microbiology and the regions where impetigo is prevalent.
[#Response#]: Thanks, and we have now moved the contents from line #83–98 under ‘Microbiology of impetigo’ title as suggested. (Please see Page #2 and Line #67–76)
- [Comment*]: I also would suggest to start a new section titled Clinical presentation. (Line #64 – Line#82)
[#Response#]: Thanks, and we have now created a new subsection or subtopic titled with “clinical presentation” and presented all the contents from line #64–82 under this subtopic as suggested. (Please see Page #2–3 and Line #77–99)
- [Comment*]: Line #64: Delete the “forms” after non-bullous.
[#Response#]: Thanks, and the word “forms” is now deleted. (Please see Page #2 and Line #78)
- [Comment*]: Line #74: I don’t think it is necessary to repeat the causative organism since you have already reported the common bacteria/organism causing impetigo under microbiology of impetigo.
[#Response#]: Thanks. The presentation of causative agents under “microbiology of impetigo” is for impetigo in general. However, the presentation of causative agents under the part of interest is specific to the different clinical presentations of impetigo such as bullous and non-bullous forms. Bullous form is mainly caused by S. aureus while non-bullous is caused by either S. aureus alone or GAS alone or a combination of two. This is the overall notion of this part.
- [Comment*]: You discuss the sites where non-bullous impetigo is commonly seen, if know, consider reporting the sites on bullous impetigo. In addition, it sounds like the non-bullous impetigo is classifies as primary and secondary, please verify.
[#Response#]: Thanks, and this part is now modified including the areas where bullous impetigo commonly occurs as follows “Bullous impetigo clinically presents with large fragile, flaccid, and fluid filled bullae around the trunk and upper extremities that less readily rupture into thin, brown crusts, and result from exfoliative toxins produced by S. aureus species (Figure 1) [4, 12, 20].” (Please see Page #2 and Line #79–81)
Regarding non-bullous impetigo classified as primary and/or secondary impetigo, the classification of impetigo into primary and secondary infection is based on the pathway through which the bacteria enter the skin and cause the infection, and not attributed to the clinical presentation. This is not specific for bullous impetigo–meaning it works for both bacteria. Also, it is not our intention to state that this classification is exclusive to non-bullous impetigo.
- [Comment*]: Line #83: Needs clarification, I think the presentation should discuss the Bullous and non-bullous
[#Response#]: Thanks, and we have already explained the clinical appearance or presentations of both forms in text and figure under ‘clinical presentation’ subtopic. (Please see Page #2–3 and Line #77–99)
- [Comment*]: Line #84: Correct the wording “both forms appear to be similar”.
[#Response#]: Thanks, and the phrase “is appeared to be similar” is corrected as “appear to be similar”. (Please see Page #3 and Line #98–99)
- [Comment*]: Line #98: Delete “isolates” at the end of the sentence. You have mentioned this.
[#Response#]: Thanks, and the word “isolates” is now deleted. (Please see Page #2 and Line #76)
Impetigo Morbidity
- [Comment*]: Line #100: Should this say “morbidity from impetigo, instead “of”?
[#Response#]: Thanks, and this part is now modified as “Morbidity from impetigo”. (Please see Page #3 and Line #100)
- [Comment*]: Line #111 & Line #124: Correct “ASPGN” to “APSGN”
[#Response#]: Thanks, and “ASPGN” is now corrected as “APSGN” for both cases. (Please see Page #3 and Line #112 and Page #4 and Line #125)
- [Comment*]: Line #129: Delete “more than 17 times” and replace with “much higher”, the numbers in the parenthesis clarify the numbers.
[#Response#]: Thanks, and “more than 17 times” is now replaced with “much higher”. (Please see Page #4 and Line #130)
- [Comment*]: Line #133: Delete the second “after” and replace with 2-3 weeks post streptococcal infection.
[#Response#]: Thanks, the phrase “after the” is now deleted and replaced with “post”. (Please see Page #4 and Line #134)
- [Comment*]: Line #136 – Line #140: Long sentence and confusing. You report that well-designed studies are yet requires, but then mention about reports indicating high rates: where are these reports coming from?
[#Response#]: Thanks, and this part is now modified as “There is a lack of well-designed studies to establish a clear link between impetigo and ARF in Indigenous communities [33]. However, reports from Australia, and New Zealand [34-36] indicating high rates of ARF in settings with high incidence of impetigo and low rates of streptococcal pharyngitis could provide some insight into a plausible link between impetigo and ARF and its subsequent progression to RHD”. (Please see Page #4 and Line #137–142)
- [Comment*]: Line #143 and Line #144: Not sure if the exact numbers of 67 times or 64 times is necessary, I think you can just say more likely to develop ARF or have RHD than…..
[#Response#]: Thanks, and we included these figures to clearly show the real burdens of the ARF and RHD in Aboriginal children as opposed to their non-Aboriginal counterparts. Hence, we would like to keep these representations accordingly.
- [Comment*]: Line #149: Change “found” to “characterized” by pain….
[#Response#]: Thanks, the phrase “found with” is now replaced with “characterized by”. (Please see Page #4 and Line #150)
- [Comment*]: Line #150: Change “lack of sleep” to “sleep disturbances” impacting the wellbeing of those affected.
[#Response#]: Thanks, and this part is now modified as “Apart from these potential chronic complications, impetigo is typically characterized by pain, itching, discomfort, and sleep disturbance, substantially impacting the wellbeing of those affected [40].” (Please see Page #4 and Line #150–152)
- [Comment*]: Line #151 – Line #154: Should this say “due to the contagious nature of this infection, often the children are forced to stay home from school or daycare until the infection resolves. This in turn may require that the parents have to take time away from work to care for the children.
[#Response#]: Thanks, and this part is now modified as “Due to the contagious nature of the infection, the children are often forced to stay home, excluded from schools or daycares until the infection resolves, and this in turn may require the parents to take time away from work to care for their children [5, 41].” (Please see Page #4 and Line #152–154)
CURRENT IMPETIGO TREATMENT & CHALLENGES
- [Comment*]: Line #156 & Line #157: Two short sentences which can be combined for better flow. Here is a suggestion. “Antibiotic therapy is indicated for faster symptom resolution as well as eliminating and/or limiting the spread of the disease from person-to-person”.
[#Response#]: Thanks, and this part is now modified as “Antibiotic therapy is indicated for faster symptom resolution as well as eliminating and/or limiting the spread of the disease from person-to-person [4, 17, 21].” (Please see Page #4 and Line #156–157)
- [Comment*]: Line #157 – Line #160: Just like above, could be shorter and less repetitive. In Australia, mild or moderate impetigo is usually treated with topical antibiotics (i.e., mupirocin) whereas severe or recurrent infections are treated with oral antibiotics (i.e., …….).
[#Response#]: Thanks, and this part is now modified as “In Australia, mild and moderate impetigo forms are usually treated with topical antibiotics (i.e. mupirocin) whereas severe or recurrent infections, are treated with oral antibiotics (i.e. dicloxacillin, flucloxacillin, cefalexin and trimethoprim/sulfamethoxazole [co-trimoxazole]) [42].” (Please see Page #4 and Line #157–160)
- [Comment*]: Line #161 – Line #167: Change and simplify to, “several systematic reviews suggest topical therapy for first line and systemic therapy for extensive infection. However, there is a lack of clear conclusive evidence between the efficacy of both treatments. Further, a significant number of reports suggest that topical treatments may be superior or even equivalent to oral therapy even for the treatment of extensive impetigo infection.
[#Response#]: Thanks, and this part is now modified as “Several systematic reviews also suggest topical antibiotics as the treatment of choice for impetigo and systemic antibiotics for complicated impetigo with extensive infections [13, 17, 43, 44]. However, there is a lack of clear conclusive evidence regarding the difference in clinical efficacy between topical and systemic antibiotics for impetigo [13]. A significant number of reports suggest that topical treatments may be superior or equivalent to the oral therapy even for treating extensive form of impetigo [17, 21, 45-48].” (Please see Page #4 and Line #160–165)
- [Comment*]: Line #168 & Line#169: Delete the second “systemic” just say side effects, in addition, delete the whole content in the () and just say i.e. Gastrointestinal.
[#Response#]: Thanks, and this part is now modified as per the suggestion as “Unlike systemic antibiotics, topical therapies reduce the potential for systemic absorption and side effects (i.e. gastrointestinal), and also lower the potential for developing resistance to life saving systemic antibiotics [11, 13, 17, 46].” (Please see Page #4 and Line #166–168)
- [Comment*]: Line #173: Delete “at the site of action”, you just said they are applied to the infection site.
[#Response#]: Thanks, and the phrase “at the site of action” is now deleted. (Please see Page #5 and Line #171)
- [Comment*]: Line #175: Again, you have two “treatments” here, delete one.
[#Response#]: Thanks, and the second “treatment” is now deleted. (Please see Page #5 and Line #172)
- [Comment*]: Line #176 – Line #180: Needs revision and cleaning. Not sure how to help. Very confusing and repetitive.
[#Response#]: Thanks, and this part is now modified as “Like other antibiotics, overuse of topical antibiotics can drive increased antimicrobial resistance (AMR) and rapid emergence of multidrug-resistant bacterial strains (MRSA and macrolide-resistant S. pyogenes) is essentially threatening the availability of life saving treatments [16, 18, 46, 50, 51]”. (Please see Page #5 and Line #174–176)
- [Comment*]: Line #181: The word “priority” does not fit here. I think “widespread” pathogen may be better.
[#Response#]: Thanks, and this term is adopted from WHO document and we have now put it in quotation as “priority pathogen”. (Please see Page #5 and Line #177)
Please also see the following link for more on this document: https://www.who.int/news/item/27-02-2017-who-publishes-list-of-bacteria-for-which-new-antibiotics-are-urgently-needed
- [Comment*]: Line #193: Is this supposed to be in the near future?
[#Response#]: Thanks, and it is corrected as “in the near future”. (Please see Page #5 and Line #189)
- [Comment*]: Line #198: Change “front-line” to “first-line”.
[#Response#]: Thanks, the word “front-line” is now replaced with “first-line”. (Please see Page #5 and Line #195)
- [Comment*]: Line #200: Change “adverse consequences” to “adverse effects” as above to keep it consistent. Delete the side effects in the ().
[#Response#]: Thanks, and this part is now modified as “…they are also associated with adverse effects (particularly gastrointestinal)”. (Please see Page #5 and Line #197–198)
- [Comment*]: Line #201 – Line #204: Just avoid too much wording and say “In addition, given the rapid emergence of CA-MRSA in remote areas,
[#Response#]: Thanks, and this part is now modified as “In the long run, given the rapid emergence of community-associated methicillin-resistant S. aureus (CA-MRSA) in remote areas….”. (Please see Page #5 and Line #198–199)
- [Comment*]: Line #204 & Line #205: delete this sentence not necessary. You have a nice section following with the “based on a 7-year…..
[#Response#]: Thanks, and the sentence of interest has now been deleted. (Please see Page #5 and Line #201)
- [Comment*]: Line #215: What is “post-antibiotic world is fast becoming a reality” mean?
[#Response#]: Nowadays, it is quite clear that microbial resistance is emerging faster than the pace of replacing the existing treatments with new antimicrobial agents, severely limiting the options available to treat increasingly resistant infections. Given the seriousness of AMR, the World Health Organization and the research community have recently highlighted the critical need for research into the resistance patterns of pathogens to the current antibiotics and the development of new and alternative antimicrobial agents for treating infections caused by these resistant bacteria. Without decisive, urgent and concerted actions, we are heading for a post-antibiotic era, a time when our antibiotics will become useless and common infections and minor injuries can once again kill people. Hence, this is the reason we used the phrase “post-antibiotic world is fast becoming a reality”.
To make this conclusively clear, we have now modified this part as “Given the rapid emergence of these resistant bacteria to the current topical antibiotics, a post-antibiotic era is fast approaching, requiring alternative means of treatments with an intension to break the cycle, and prevent further resistance.” (Please see Page #5 and Line #210–212)
- [Comment*]: Line #233 – Line #234: A different way of reporting will be, this in turn requires using the co-trimazole tablets to compound the syrup which not only has intolerable taste (I thought for pediatric compounding various sweetening agents are available) but also does not meet the regulatory standards specially for accurately administering to children. “Obnoxious taste” is not appropriate vocabulary here.
[#Response#]: Thanks, and there is no information whether the health professionals in those settings compound the tablets to a syrup form. The sentence of interest refers to the practice of breaking the adult dose tablets and administering to children. We have now replaced the phrase “obnoxious test” with “intolerable taste” and modified the sentence as per the suggestion.
The modified version now reads as “This, in turn, has left the health professionals in the area with the alternative option of crushing and giving the adult dose of co-trimoxazole tablets to children [73]. This practice, not to mention the intolerable taste of the crushed tablets, is not recommended by antibiotic regulators as it does not meet the regulatory standards for administering a dose accurately to young children [73].” (Please see Page #6 and Line #220–224)
- [Comment*]: Line #236 – Line #241: Similarly, with BPG suspension there are several limitations: first, the volume of injection (); second, the viscosity (); third, irritant nature (), results in painful administration which often leads to poor compliance.
[#Response#]: Thanks, and this part is now modified as “Similarly, BPG administration is painful and often leads to poor compliance – this is due to the higher injection volume (1.6–2.3ml per injection), viscosity (due to high concentration of suspended BPG particles) and irritant nature of the suspension (leading to injection site reaction such as pain, inflammation, erythema, swelling, and skin ulcer) [39, 73, 74].” (Please see Page #6 and Line #225–228)
- [Comment*]: Line #241 – Line #242: In addition, in the case of recurrent impetigo infection, the fear of painful injection may result in needle phobia and non-compliance with therapy.
[#Response#]: Thanks, and this part is now modified as “In addition, in case of recurrent impetigo infection, the fear of painful injection may result in needle phobia and non-compliance with the therapy [74].” (Please see Page #6 and Line #228–230)
- [Comment*]: Line #244: Consider this change: often results not only in poor compliance but also suboptimal clinical outcome”.
[#Response#]: Thanks, and this part is now modified as “This shows that the current impetigo treatment options for Indigenous impetigo patients seem far more challenging that it often results in suboptimal clinical outcomes” (Please see Page #6 and Line #230–231)
- [Comment*]: Line #246-Line #252: Impetigo repeated four times. Please consider deleting the one on Line #250.
[#Response#]: Thanks, and one of the “impetigo” words is deleted as suggested. (Please see Page #6 and Line #236)
- [Comment*]: Line #255: Change “alternative antimicrobial” to “alternative selection”. Delete the candidates.
[#Response#]: Thanks, and “alternative antimicrobial candidates” is now modified as “alternative selections”. (Please see Page #6 and Line #242)
POTENTIAL ANTIMICROBIAL CANDIATES FOR IMPETIGO
Tee Tree oil
- [Comment*]: Line #265: Delete “of” before microorganisms.
[#Response#]: Thanks, and “of” is now deleted as suggested. (Please see Page #6 and Line #251)
- [Comment*]: Line #269: Replace “in the use of essential oils” with “interest in their use as substitutes for...
[#Response#]: Thanks, and “of essential oils” is replaced with “their”. (Please see Page #6 and Line #256)
- [Comment*]: Line #273: “Exhibit” should be ”exhibits”.
[#Response#]: Thanks, and “exhibit” is now corrected as “exhibits”. (Please see Page #6 and Line #260)
- [Comment*]: Line #278: Change “skin application” to “topical application.
[#Response#]: Thanks, and “skin” is now changed to “topical”. (Please see Page #7 and Line #265)
- [Comment*]: Line #283: Change “admiration” to “interest”.
[#Response#]: Thanks, and “admiration” is now changed to “interest”. (Please see Page #7 and Line #270)
- [Comment*]: Line #288: Not sure if I saw the definition of “MBC”. Please define ”minimum bactericidal concentration”?
[#Response#]: Thanks, and both “MIC” and “MBC” have now been defined as suggested. (Please see Page #7 and Line #275–276)
- [Comment*]: Line #291: Replace “attracted” to “gained”.
[#Response#]: Thanks, and the word “attracted” is replaced with “gained”. (Please see Page #7 and Line #279)
- [Comment*]: Line #294: Revise to, allocating MRSA infected patients (n=30) to either routine care () or TTO (). Delete the words “group” not needed.
[#Response#]: Thanks, and this part has now been modified as “Caelli et al (2000) evaluated the clinical efficacy of TTO by randomly allocating MRSA infected patients (n=30) to either routine care (2% mupirocin nasal ointment and triclosan body wash, no report on dose and frequency of administration) or TTO (a 4% tea tree oil nasal ointment and 5% tea tree oil body wash, no report on dose and frequency of administration) given for a minimum of three days [89].” (Please see Page #7 and Line #281–285)
- [Comment*]: Line #305: Delete “received the treatments”.
[#Response#]: Thanks, and “received the treatments” is now deleted as suggested. (Please see Page #7 and Line #292)
- [Comment*]: Line #307: 47% vs. 31% (Vs?)
[#Response#]: Thanks, and this part has now been modified as “TTO treatment was also highly effective at clearing superficial skin lesions compared to the standard treatment (47% versus 31%, respectively, no report on P values), indicating its potential use for MRSA-implicated skin infections like impetigo.”. (Please see Page #7 and Line #294–297)
- [Comment*]: Line #319: Consider changing “skin diseases” to Skin infections”.
[#Response#]: Thanks, and “skin diseases” is now changed to “skin infections”. (Please see Page #7 and Line #307)
Manuka Oil
- [Comment*]: Line #338: Delete “and for” and say “an analgesic, and wound….
[#Response#]: Thanks, and “and for” is now replaced with “an”. (Please see Page #8 and Line #325–326)
- [Comment*]: Line #344: Delete “s” from cosmetic. It should just be “cosmetic products”.
[#Response#]: Thanks, and “s” is now deleted. (Please see Page #8 and Line #332)
- [Comment*]: Line # 349: Change “skin disease” to “skin infections or skin conditions” for consistency.
[#Response#]: Thanks, and “skin diseases” is now changed to “skin infections”. (Please see Page #8 and Line #337)
Hydrogen Peroxide
- [Comment*]: Line #381: Change “frontline” to “first line”.
[#Response#]: Thanks, and “frontline” is now changed to “first-line”. (Please see Page #9 and Line #370)
SUMMARY
- [Comment*]: “Impetigo” has been repeated multiple times in this
[#Response#]: Thanks, and we have attempted to minizine the number of “impetigo” words in this part. (Please see Page #9 and Line #372–386)
- [Comment*]: Line 384 and Line #388 – The term “frontline” is not appropriate for the content. Use first-line as it has been listed in previous section(s) of the manuscript.
[#Response#]: Thanks, and “frontline” is now changed to “first-line”. (Please see Page #9 and Line #378)
- [Comment*]: Line #385: The term “demanding” does not flow here. Consider “requiring”
[#Response#]: Thanks, and “demanding” is now changed to “requiring”. (Please see Page #9 and Line #380)
- [Comment*]: Line #386: What do you mean by “high-density settings”?
[#Response#]: Thanks, and “high-density settings” is now changed to “endemic settings”. (Please see Page #9 and Line #381)
- [Comment*]: Line #389: Delete “precious”, not necessary and adds no meaning…
[#Response#]: Thanks, and “precious” is now deleted. (Please see Page #9 and Line #384)
- [Comment*]: Line #392: Financial support: change “is” to “was”.
[#Response#]: Thanks, and “is’ is now changed to “was”. (Please see Page #9 and Line #387)
FIGURES:
- [Comment*]: Not sure if Figure 4 is necessary
[#Response#]: Thanks, and “figure 4” is now deleted. (Please see Page #5 and Line #212–213)
REFERENCES:
- [Comment*]: No comments, hope it is appropriate per Journals guidelines.
[#Response#]: Thanks.
Reviewer 2 Report
Dear Authors,
The article “Intolerable burden of impetigo in endemic settings: A review of “state-of-play” and future directions for research on alternative treatments” is very interesting work, reporting the study connected with the new and effective impetigo treatments options. The article underlines the importance of the newer topical treatment alternatives derived from herbal medicine, especially putting attention to strong antimicrobial activity of tea tree and manuka essential oils, which are proposed as natural substitutes for synthetic antimicrobials. As well as well-known antiseptic agent - hydrogen peroxide. The Authors made an extensive literature review on these substances, in which they collected relevant information about their strong antimicrobial activity against S. aureus, GAS and their resistant isolates (low MIC and MBC values). It is worth noting the great potential of tea tree oil as a potent anti-biofilm formation agent in case of MRSA strains with good clinical safety profile and low chance for developing resistance.
The manuscript is interesting, is written in a correct English and reports promising findings based on properly selected, 132 scientific reports. The article structure is appropriate. However, it requires some minor corrections before acceptance, such as improvement and extension of conclusions. Summing up, I recommend to publish this article after checking for possible grammar/spelling typos and after considering the comments in minor concerns.
Minor concerns:
- Please adapt the citation style in accordance to Journal requirements [in square brackets]
- Please expand the conclusion section.
- Check for possible grammar/spelling typos
- Line “4” of the manuscript, there is a mistake with “and PhD”
Kind regards
Author Response
Reviewer #2
GENERAL COMMENT
The article “Intolerable burden of impetigo in endemic settings: A review of “state-of-play” and future directions for research on alternative treatments” is very interesting work, reporting the study connected with the new and effective impetigo treatments options. The article underlines the importance of the newer topical treatment alternatives derived from herbal medicine, especially putting attention to strong antimicrobial activity of tea tree and manuka essential oils, which are proposed as natural substitutes for synthetic antimicrobials. As well as well-known antiseptic agent - hydrogen peroxide. The Authors made an extensive literature review on these substances, in which they collected relevant information about their strong antimicrobial activity against S. aureus, GAS and their resistant isolates (low MIC and MBC values). It is worth noting the great potential of tea tree oil as a potent anti-biofilm formation agent in case of MRSA strains with good clinical safety profile and low chance for developing resistance.
The manuscript is interesting, is written in a correct English and reports promising findings based on properly selected, 132 scientific reports. The article structure is appropriate. However, it requires some minor corrections before acceptance, such as improvement and extension of conclusions. Summing up, I recommend to publish this article after checking for possible grammar/spelling typos and after considering the comments in minor concerns.
[#Response#]: We are thankful to reviewer #2 for the kind words and comments
MINOR CONCERNS:
- [Comment*]: Please adapt the citation style in accordance to Journal requirements [in square brackets
[#Response#]: Thanks, and we have now modified the citation style as suggested. (Please see Page #1–9)
- [Comment*]: Please expand the conclusion section.
[#Response#]: Thanks, and we have now expanded the conclusion part. (Please see Page #9 and Line #372–386)
- [Comment*]: Check for possible grammar/spelling typos
[#Response#]: Thanks, and we have checked for possible grammar/spelling typos as suggested.
- [Comment*]: Line “4” of the manuscript, there is a mistake with “and PhD”
[#Response#]: Thanks, and we have now corrected this as suggested. (Please see Page #1 and Line #4)